health and disease and epidemiology, microbiology, genetics

morbillivirus, virology, viral genetics, seal, unusual mortality event, wildlife disease

**Authors for correspondence:**
Wendy Puryear
e-mail: wendy.puryear@tufts.edu
Kaitlin Sawatzki
e-mail: kaitlin.sawatzki@tufts.edu

†These authors contributed equally to this study.

# Longitudinal analysis of pinnipeds in the northwest Atlantic provides insights on endemic circulation of phocine distemper virus

Wendy Puryear[1,†], Kaitlin Sawatzki[1,†], Andrea Bogomolni[2], Nichola Hill[1], Alexa Foss[1], Iben Stokholm[3], Morten Tange Olsen[3], Ole Nielsen[4], Thomas Waltzek[5], Tracey Goldstein[6], Kuttichantran Subramaniam[5], Thais Carneiro Santos Rodrigues[5], Manjunatha Belaganahalli[7], Lynda Doughty[8], Lisa Becker[9], Ashley Stokes[10], Misty Niemeyer[11], Allison Tuttle[12], Tracy Romano[12], Mainity Batista Linhares[13], Deborah Fauquier[14] and Jonathan Runstadler[1]

[1]Department of Infectious Disease and Global Health, Cummings School of Veterinary Medicine, Tufts University, North Grafton, MA, USA
[2]Department of Marine Science, Safety and Environmental Protection, Massachusetts Maritime Academy, Buzzards Bay, MA, USA
[3]Evolutionary Genomics Section, GLOBE Institute, University of Copenhagen, Denmark
[4]Department of Fisheries and Oceans Canada, Winnipeg, Canada
[5]Department of Infectious Diseases and Immunology, College of Veterinary Medicine, University of Florida, Gainesville, FL, USA
[6]Karen C. Drayer Wildlife Health Center and Department of Pathology, Immunology and Microbiology, School of Veterinary Medicine, University of California, Davis, CA, USA
[7]Institute of Animal Health and Veterinary Biologicals Karnataka Veterinary, Animal and Fisheries sciences University Hebbal, Bengaluru, India
[8]Marine Mammals of Maine, Bath, ME, USA
[9]National Marine Life Center, Buzzards Bay, MA, USA
[10]Seacoast Science Center, Rye, NH, USA
[11]International Fund for Animal Welfare, Yarmouth Port, MA, USA
[12]Mystic Aquarium, Mystic, CT, USA
[13]School of Veterinary Science, University of Queensland, Brisbane, Australia
[14]Office of Protected Resources, National Marine Fisheries Service, National Oceanic and Atmospheric Administration, Silver Spring, MD, USA

WP, 0000-0002-1795-3399; KS, 0000-0002-7933-3064; AB, 0000-0002-1989-023X; NH, 0000-0003-3213-6693; IS, 0000-0003-4097-251X; MTO, 0000-0001-6716-6345; TW, 0000-0001-6383-5978; TG, 0000-0002-1672-7410; KS, 0000-0002-2752-4467; TCSR, 0000-0001-6992-1295; MB, 0000-0003-3401-5799; MBL, 0000-0003-4988-6984; JR, 0000-0002-6747-7765

Phocine distemper virus (PDV) is a morbillivirus that circulates within pinnipeds in the North Atlantic. PDV has caused two known unusual mortality events (UMEs) in western Europe (1988, 2002), and two UMEs in the northwest Atlantic (2006, 2018). Infrequent cross-species transmission and waning immunity are believed to contribute to periodic outbreaks with high mortality in western Europe. The viral ecology of PDV in the northwest Atlantic is less well defined and outbreaks have exhibited lower mortality than those in western Europe. This study sought to understand the molecular and ecological processes underlying PDV infection in eastern North America. We provide phylogenetic evidence that PDV was introduced into northwest Atlantic pinnipeds by a single lineage and is now endemic in local populations. Serological and viral screening of pinniped surveillance samples from 2006 onward suggest there is continued circulation of PDV outside of UMEs

among multiple species with and without clinical signs. We report six full genome sequences and nine partial sequences derived from harbour and grey seals in the northwest Atlantic from 2011 through 2018, including a possible regional variant. Work presented here provides a framework towards greater understanding of how recovering populations and shifting species may impact disease transmission.

# 1. Introduction

Phocine distemper virus (PDV) is a phocine morbillivirus in the *Morbillivirus* genus that has been associated with four unusual mortality events (UMEs) among pinnipeds in the North Atlantic, including two in western Europe (northeast Atlantic) and two in eastern North America (the northwest Atlantic). In all events, harbour seals (*Phoca vitulina*) were the primary impacted species, with some inclusion of grey seals (*Halichoerus grypus)*. Both UMEs that occurred in Europe had significant mortality, while those in North America had relatively mild observable impacts to the population, though deaths did occur. There have also been scattered reports of PDV-positive animals between UMEs in North America, while PDV outside of UMEs has rarely been detected in Europe, despite annual surveillance [1,2].

The first known PDV epizootic occurred in Europe in 1988 and killed an estimated 23 000 harbour seals over less than a year [3,4]. This outbreak was attributed to a massive yet transient incursion of harp seals from the Arctic, shortly predating the UME [5,6]. Another northeast Atlantic outbreak occurred in 2002 following a similar geographical and temporal spread, killing an estimated 30 000 harbour seals, potentially 60% of the regional population [4,7,8]. There has been very little evidence for the presence of PDV within Europe prior to or outside of the two UMEs, with cumulative sampling effort including approximately 1000 animals in German and Danish waters, spanning 25 years after the first UME. While there were seropositive animals in the years immediately following the 1988 and 2002 outbreaks, population immunity rapidly declined with no evidence of sustained exposure [2,9–11].

In contrast to the distinct and punctuated events in Europe with close temporal association to UMEs, the history of PDV in eastern North America has been less clear. Clinical signs consistent with PDV were first observed in stranded seals along the northwest Atlantic in 1991, though serology suggests regional undocumented or subclinical infections may have been present as early as 1981 [12,13]. Despite its apparent presence in the region, the first UME did not occur in North America until 2006, and then again in 2018 [14–16]. When both of these UMEs occurred, the mortality was much less significant than that seen in Europe and is estimated to have only affected 2% of the northwest Atlantic harbour and grey seal population [17]. The impacted species are found on both sides of the Atlantic, and there is evidence of minimal genetic variation in PDV derived from the first three outbreaks (1988, 2002, 2006). Therefore, we sought to understand factors contributing to the differences in viral ecology following introduction of a pathogen [4,7,18,19].

Harp seals (*Pagophilus groenlandicus*) local to the Barents Sea have been proposed as the primary PDV reservoir due to an unusual mass population movement into the North Sea preceding the 1988 outbreak. This is supported by contemporaneous serology demonstrating antibodies in the population [5,6,20]. In a retrospective analysis, Dietz *et al.* [12] found that 30% (12/40) harp seals on the western coast of Greenland in 1985–1986 had antibodies against canine distemper virus (CDV) and suggested these were likely to be PDV antibodies. However, only the 1988 northeast Atlantic PDV outbreak was preceded by unusual southern harp seal migrations and the proximal causes of the 2002, 2006 and 2018 outbreaks are unclear. These data suggest that while harp seals play an important role as reservoirs, there remain complex, undefined factors that contribute to PDV outbreaks.

The logistical challenges of conducting systemic surveillance in marine mammals have meant that the ability to define temporal and spatial dynamics of PDV has remained elusive. There are significant data gaps, very little prevalence data and minimal available sequences, particularly from North America. The dynamics of PDV transmission and evolution have direct impacts on pinniped health, but also provide an important template for looking at the ways in which a changing Arctic and shifting populations and their pathogens can impact health across different species [21]. Declines in Arctic sea ice and subsequent shifts in habitat use have been associated with the recent detection of PDV in multiple marine mammal species in the Pacific Northwest for the first time [22,23].

This study represents the first analysis of PDV in North American seals from the first UME in 2006 through 2020 and includes data from harbour, grey and harp seals. By leveraging archived samples, ongoing data collections for health assessments, and samples from the most recent UME in 2018, we provide a significant increase in the number of full genome PDV sequences, and the number of partial sequences from North America. Using phylogenetics supported by serology and detection of viral RNA, we reconstruct the evolutionary history of PDV in North America in correlation to species distribution, population immunity and viral variation. Finally, the work presented here provides a framework to understand the distinct outcomes that characterize PDV outbreaks in Europe as compared to North America.

# 2. Methods

## (a) Sample collection

Samples were obtained from grey, harbour and harp seals from 2010 through 2020. Samples were collected from wild and stranded animals, and included swabs, sera and tissue. See electronic supplementary material, table S1 for a summary of species, year and sample type.

### (i) Live wild pup samples

Swab samples were collected from 492 weaned wild-caught grey seal pups in the Gulf of Maine during the months of January and February from 2016 to 2020 on rookeries at Muskeget Island, Monomoy Island, Great Point Nantucket and Seal Island. Pup captures and sampling were performed as described in Puryear *et al.* [24]. Conjunctival, nasal and rectal samples were collected on polyester tipped swabs (Puritan Medical, Guilford, ME) and transported on cold chain in viral transport media (VTM) (M4RT from Fisher Scientific, Hampton, NH). Pup length and girth were used to calculate body condition index (BCI) as BCI = (girth ÷ length) × 100 [25,26].

### (ii) Stranded animal samples

Conjunctival, nasal and/or rectal samples were collected at the National Marine Life Center, International Fund for Animal

Welfare (IFAW), Mystic Aquarium, Seacoast Science Center, Marine Mammals of Maine (MMoME) and National Aquarium from both live and dead pinnipeds that stranded along the Eastern Atlantic seaboard spanning Maine to Maryland, from 2018 to 2020. Samples from stranded animals were collected as diagnostic samples under each organizations Stranding Agreement with NOAA Fisheries Service. Tissue samples were obtained from necropsies performed by MMoME or IFAW on a subset of harbour ($n = 12$) and grey ($n = 9$) seals that stranded in Maine or Massachusetts from 2011 through 2018 with clinical signs consistent with PDV [27]

## (b) Serology

Sera was obtained from 283 pinnipeds and tested for the presence of PDV antibodies using a monoclonal antibody-based competitive enzyme linked immunosorbent assay; titres 8 and above were recorded as positive [28]. Rehabilitated pinniped samples from the Marine Animal Research Center/University of New England were sent to Oklahoma Animal Disease Diagnostic Laboratory or the Marine Mammal Diagnostic Service Laboratory (University of Georgia, Athens, GA). Adult wild capture harbour and grey seal serology samples were processed at the University of Georgia.

## (c) RNA extraction

### (i) Swabs

RNA was extracted from swabs as described in Puryear *et al.* [24]. Briefly, samples were vortexed and 50 µl aliquoted for semi-automated extraction using the Mag-Bind Viral DNA/RNA extraction kit (Omega, Norcross, GA) and Kingfisher platform (ThermoFisher).

### (ii) Tissue

Lung and brain tissue from 2018 were homogenized with 2.8 mm stainless steel grinding balls. RNA was extracted using cold RNAzol RT (MRC, Cincinnati, OH) or QIAamp Viral RNA Mini Kit (Qiagen, Valencia, CA) as per the manufacturer instructions. RNA was extracted from tissues harvested in 2011–2015 using methods described by Bogomolni *et al.* [29].

## (d) Real-time RT-PCR PDV screening

RNA was screened for the PDV hemagglutinin (H) gene by reverse transcription polymerase chain reaction (RT-PCR) on the StepOne-Plus platform (ABI, Beverly, MA) using qScript XLT One-Step RT-qPCR ToughMix ROX (VWR, Franklin, MA) and the previously described PDV H 116 primer set [29]. RT-PCR was performed as follows: 50°C, 10 min; 95°C, 1 min; (95°C, 3 s; 60°C, 30 s) for 45 cycles.

## (e) Sequence generation

Full viral genome and hemagglutinin genes were sequenced from raw sample-derived RNA using next-generation sequencing platforms Illumina (HiSeq, MiSeq) and Oxford Nanopore Technologies (MinION). Complete methods describing library preparation, sequencing kits and assembly are detailed in electronic supplementary material, methods. GenBank accession numbers are as follows: full genome—MW642077 (US/Hg/IFAW18-001/2018), MW642078 (US/Pv/NEAQ18-044/2018), MW504062 (CA/Pg/WVL181140/2017), MW581015 (US/Pv/MME-287/2018), MW581016 (US/Pv/MME-343/2018), MW581017 (US/Pv/MME-430/2018); hemagglutinin—MW581018–MW581026.

## (f) Phylogenetics

### (i) Time-scaled phylogenetic analysis

Sequences were aligned using ClustalΩ (v.1.2.4). Genomes ($n = 52$) comprised the six full-length genomes newly reported here, two published genomes (NL/Pv/Wad/1988 and US/Pv/2006), and 44 partial genomes spanning the phosphoprotein-matrix-fusion-hemagglutinin (P-M-F-H) genomic region [19]. H gene only comprised the dataset described above for 8 full and 44 partial genomes, with an additional 9 H gene sequences newly reported here. All sequences were time-stamped to at least the year. When exact date was not known, the date was set at the mid-point of the known month or year with uncertainty spanning the time period. Model selection and clock fit are described in electronic supplementary material, Methods (table S2 and figure S1). Final analyses were independently run five times on BEAST (v.1.10.4) with a chain length of 100 million generations using a constant growth coalescent tree prior, relaxed clocked with an uncorrelated lognormal distribution, an HKY $\gamma$ substitution model and default parameters [30]. TRACER (v.1.7.1) was used to confirm convergence and 10% burn-in removed from each chain. Trees were combined to produce the maximum clade credibility tree.

## (g) Protein modelling

Homology modelling was performed to estimate and visualize the location of observed amino acid substitutions in PDV H protein. Measles virus H (PDB 2zb5), the highest quality crystal structure available, was selected as a template and US/Pv/IFAW237/2011 imported for homology modelling in SWISS-MODEL.

# 3. Results

## (a) Northwest Atlantic pinnipeds of different species and ages are seropositive for PDV

Opportunistic serum samples from 283 harbour, grey and harp seals from 2010 to 2013 (216 harbour, 37 grey, 30 harp) were obtained from stranded and wild-caught animals along the eastern North American seaboard and tested for the presence of PDV antibodies by ELISA. All three species had detectable antibodies at equivalent proportions ($\chi^2 > 0.05$), with seropositive samples detected from 21.6% grey seals, 35.7% harbour seals and 40.0% harp seals. Seropositive samples were also detected across age class, and in each year tested (figure 1a). For this analysis, all animals identified as newborn, weanling, pup, or young of the year, were binned as 'first year' (FY) ($n = 221$), while animals identified as juvenile, sub-adult, or adult were binned as 'after first year' (AFY) ($n = 62$). Overall, of the animals sampled from 2010 to 2013 when there were no known PDV associated UMEs, 31.7% of the FY seals and 43.5% of the AFY seals were seropositive ($\chi^2 > 0.05$). It is important to note that these values are not meant to reflect seroprevalence, as the sampling was opportunistic and includes animals of varying ages and likely a mix of maternal and *de novo* antibody. Rather this data demonstrates that there were seropositive animals from all three species tested in the northwest Atlantic, in both young and adult animals, four to seven years beyond the documented 2006 UME.

## (b) PDV is detected in stranded animals with and without clinical disease

We screened swab samples (oral, conjunctival and/or rectal) from harbour ($n = 247$), grey ($n = 125$) and harp ($n = 92$) seals that were admitted for rehabilitation to the Greater Atlantic

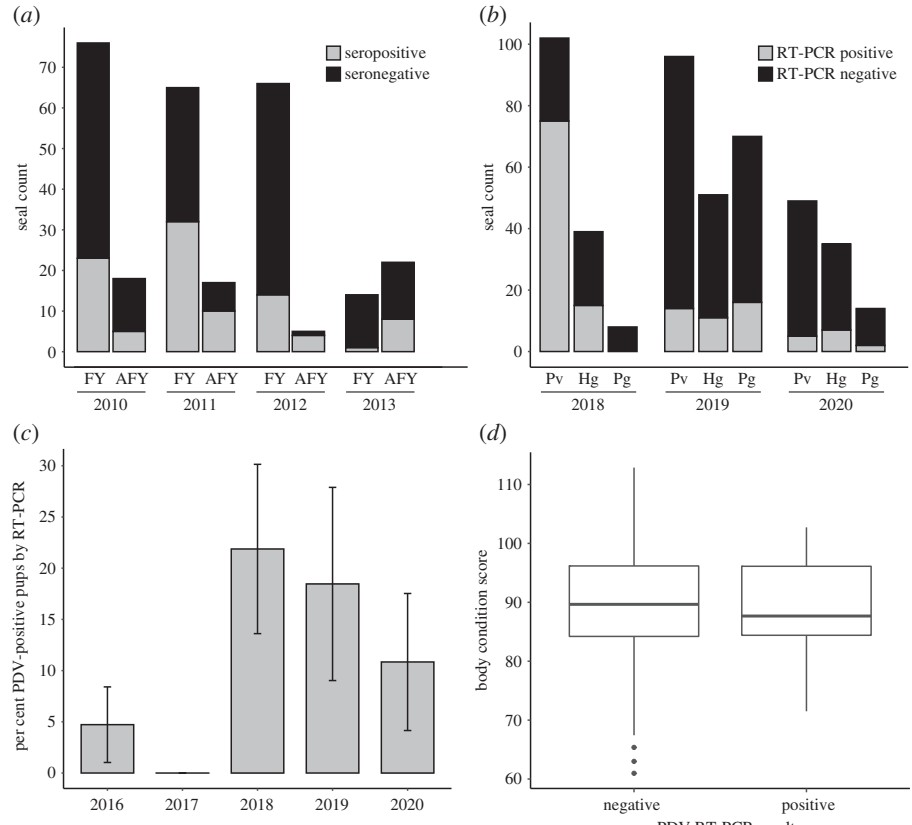

**Figure 1.** Serologic and RT-PCR evidence for ongoing PDV in the northwest Atlantic. (*a,b*) Absolute numbers are reported to reflect variation in sampling effort. (*a*) Yearly count of seropositive first-year (FY) and after first-year (AFY) animals. (*b*) RT-PCR from stranded harbour (Pv), grey (Hg) and harp (Pg) seals. (*c*) RT-PCR-positive grey seal pups wild-caught on rookeries in the Gulf of Maine, with confidence interval estimated as normal approximation to the binomial. (*d*) For each wild-caught pup, body condition was calculated as (girth ÷ length) × 100. Data were stratified based on RT-PCR-positive or negative animals; Mann–Whitney–Wilcoxon $p > 0.05$.

Region Marine Mammal Stranding Network from 2018 to 2020 for the presence of PDV by RT-PCR (figure 1*b*). During the peak of the UME in 2018, 60.4% of seals tested positive (90/149). As with prior UMEs, the proportion PCR positive for PDV was higher in harbour seals than in grey seals (73.5% and 38.5%, respectively, $\chi^2 = 0.0001$) and all (0/8) of the harp seals tested PCR negative. However, during 2019, 18.9% of stranded seals were PCR positive for PDV by RT-PCR (41/217), and 14.3% PCR positive in 2020 (14/98). In both 2019 and 2020, all three species had PDV-positive animals, with no statistically significant difference between species in either year ($\chi^2 > 0.05$).

Pinnipeds that stranded and died have sporadically been observed with clinical signs resembling PDV infection, outside of documented UMEs in the northeast US. From April 2011 through December 2015, necropsies were performed on seven harbour and eight grey seals with signs suggestive of PDV infection on Cape Cod, MA. Common gross findings in both harbour and grey seals included emphysema and meningitis. Microscopic findings in a few PCR-positive cases were suggestive of PDV infection, including pneumonia and encephalitis, but typical inclusion bodies were not identified. For each of the 15 animals, archived RNA from brain, kidney, liver, lung and/or spleen was screened. All animals were found to have at least one PDV-positive tissue by RT-PCR, primarily brain (14/15) (electronic supplementary material, table S3), confirming that sporadic PDV infection occurs in both grey and harbour seals in the northwest Atlantic outside of outbreak events.

## (c) PDV is detected on rookeries in grey seal pups without clinical disease

Grey seals haul out in dense rookery clusters during the winter pupping season, creating opportunities for viral transmission within their population [24,31]. We collected swab samples from weaned grey seal pups on four rookeries throughout the Gulf of Maine and tested for PDV. Approximately 100 animals per year were sampled during pupping seasons from 2016 to 2020 [2016 ($n = 127$), 2017 ($n = 121$), 2018 ($n = 96$), 2019 ($n = 65$), 2020 ($n = 83$)]. An animal was considered positive if one or more swabs (nasal, conjunctival, rectal) tested positive by real-time RT-PCR. Averaged across the five pupping seasons, we observed a mean prevalence of 9.76% (CI: 7.13–12.38), though significant variability was observed among years (figure 1*c*). During the two years prior to the 2018 UME, PDV was low (2.42% 2016–2017). During the 2018 pupping season immediately preceding the UME, the prevalence of PDV in pups on the rookeries was 21.88% (CI: 13.61–30.14) and then declined in 2019 (18.46%) and 2020 (10.84%). Body condition scores were not correlated with PDV status (Mann–Whitney–Wilcoxon test, $p = 0.5041$; figure 1*d*). Despite the presence of animals PCR positive for PDV on grey seal rookeries, there was no observed evidence of associated clinical signs of illness or death in pups.

### (d) Receptor binding is unlikely to present a species barrier between grey and harbour seals

To investigate potential host adaptation, we compared the viral receptor binding gene, hemagglutinin (H), sequenced from grey, harbour and harp seals. The grey ($n = 7$) and harp seal ($n = 1$) derived samples are newly reported here ($n = 7$); the harbour seal derived samples comprise those newly reported here ($n = 7$), and representative samples from 1988 to 2006 that were previously reported ($n = 6$). The putative PDV receptor binding sites from all three species were identical (electronic supplementary material, figure S2). We further performed phylogenetic analyses of the two known PDV receptors, CD150 (SLAM-F1) and Nectin-4 (PVRL-4) across a broad range of mammals. Grey and harbour seals are highly homologous for both receptors (electronic supplementary material, figures S3–S4). CD150, the probable receptor for primary infection in many tissues exhibits particularly high identity and is 99.6% identical by nucleotide and 99.7% by amino acid.

### (e) Continuous circulation of a single PDV lineage in the northwest Atlantic

Six full-length PDV genomes were sequenced from samples derived from pinnipeds from the northwest Atlantic (United States 2018: four harbour, one grey seal; Canada 2017: one harp seal [32]). These genomes have high sequence identity to published PDV genomes from 1988 to 2006 (minimum: 98.92%, maximum: 99.83%, electronic supplementary material, table S4). This dataset includes the first full genome sequences of PDV derived from a grey and harp seal. We used BEAST (v.1.10.4) to perform time-scaled phylogenetic analysis of these six new genomes, two previously published complete genomes (NL/Pv/Wad/1988, GI: 947835188; US/Pv/2006, GI: 1669600827) and the previously published complete P-M-F-H genomic region from 44 harbour seal derived samples collected during the 1988 and 2002 outbreaks in the northeast Atlantic [19]. Consistent with prior publication, these data corroborate that the two European outbreaks were separately seeded by distinct lineages (tMRCA: 1987 and 2001) with dead-end seeding events (figure 2) [18,19]. By contrast, all viruses from the northwest Atlantic fell into one clade consistent with a single incursion that occurred by 2001 (HPD 95% 1998–2002), though a more precise timeline is hindered by limited sample availability. A P-M-F-H-only alignment was highly comparable to the mixed length input (electronic supplementary material, figure S5).

### (f) A clade seen in regional pinnipeds in eastern North America exhibits a hemagglutinin substitution hotspot hypothesized to decrease fusogenic activity

PDV is a relatively stable virus and has been shown to have limited but observable mutability during outbreaks [19,33]. To understand the molecular diversity of PDV strains during and between outbreaks, we performed further analysis of H genes ($n = 61$). We selected five archived samples derived from PDV-positive harbour and grey seals (electronic supplementary material, table S3), nine harbour and grey seal derived samples from the 2018 UME (the five full genomes

described above and four additional samples), and the 2017 harp seal derived sample (full genome described above). These were combined with 46 previously published H gene sequences from the northeast and northwest Atlantic. All sequences from the northwest Atlantic spanning 2011–2015 fell into one conserved, distinct lineage that was not shared by any of the northeast Atlantic sequences with the latter being referred to as the North American lineage from here on (figure 3). While the North American lineage continued to be detected during the 2018 UME, a separate UME-specific cluster was also observed during the outbreak with only 1 differing amino acid (US/Pv/NEAQ044/2018: S24P) and six total single-nucleotide variants among all members.

When comparing the H gene sequences, we found that the North American lineage had a high number (11) of T to C substitutions in a relatively constrained location of 228 nucleotides (nt), T475C through T704C, and that these were largely translated as non-synonymous changes: F159 L, T163I, L172P, L175P, V203A, I210T and V235A (figure 4a).

The H protein forms a tetramer composed of cytoplasmic tail (CT), transmembrane (TM), stalk, connector (CO) and head domains. H protein has two forms: when it is unbound to host receptor, the head domain is down (form I) and residues involved in fusion protein (F) interactions are blocked. On receptor binding, the head domain undergoes conformational change, allowing F interaction and fusion (form II) [34]. Upon comparison and homology modelling to the homologous Measles virus H protein, we inferred the observed 228 nt region may be a substitution hotspot spanning the undefined connector region into the head domain (figure 4a). Two leucine to proline substitutions were found on the homo-dimer head interface adjacent to the connector (L172P and L175P, figure 4b). Given the location of these substitutions in a region critical for viral fusion and the persistence of this substitution over multiple years, we hypothesize viruses with this hotspot may have impaired fusogenicity.

## 4. Discussion

The data presented in this paper help to illuminate the natural history of PDV infections in the northwest Atlantic. It has been established that the first documented PDV outbreak in 1988 in Europe was strongly correlated to an unusual harp seal invasion from the Arctic into the North Sea, and that harp seals were a likely reservoir of the virus [12]. However, there were no such invasions preceding any of the other known outbreaks, nor were there major anomalous ecological events forcing abnormal species interactions [27]. It has also been an outstanding question as to why outbreaks in the northeast Atlantic seem to be more devastating than those in the northwest Atlantic. In the work presented here, we provide evidence that PDV has maintained a presence within pinnipeds of the northwest Atlantic since the initial incursion and that unlike the northeast Atlantic, overlapping species distributions and rebounding populations facilitate a sustained circulation of PDV in the northwest Atlantic.

With the additional sequence analyses described above, we estimate that PDV was introduced into the northwest Atlantic by 2001 (figures 2 and 5a), probably by harp seals either directly, or through grey seals as an intermediate. However, unlike the northeast Atlantic, once introduced, PDV had

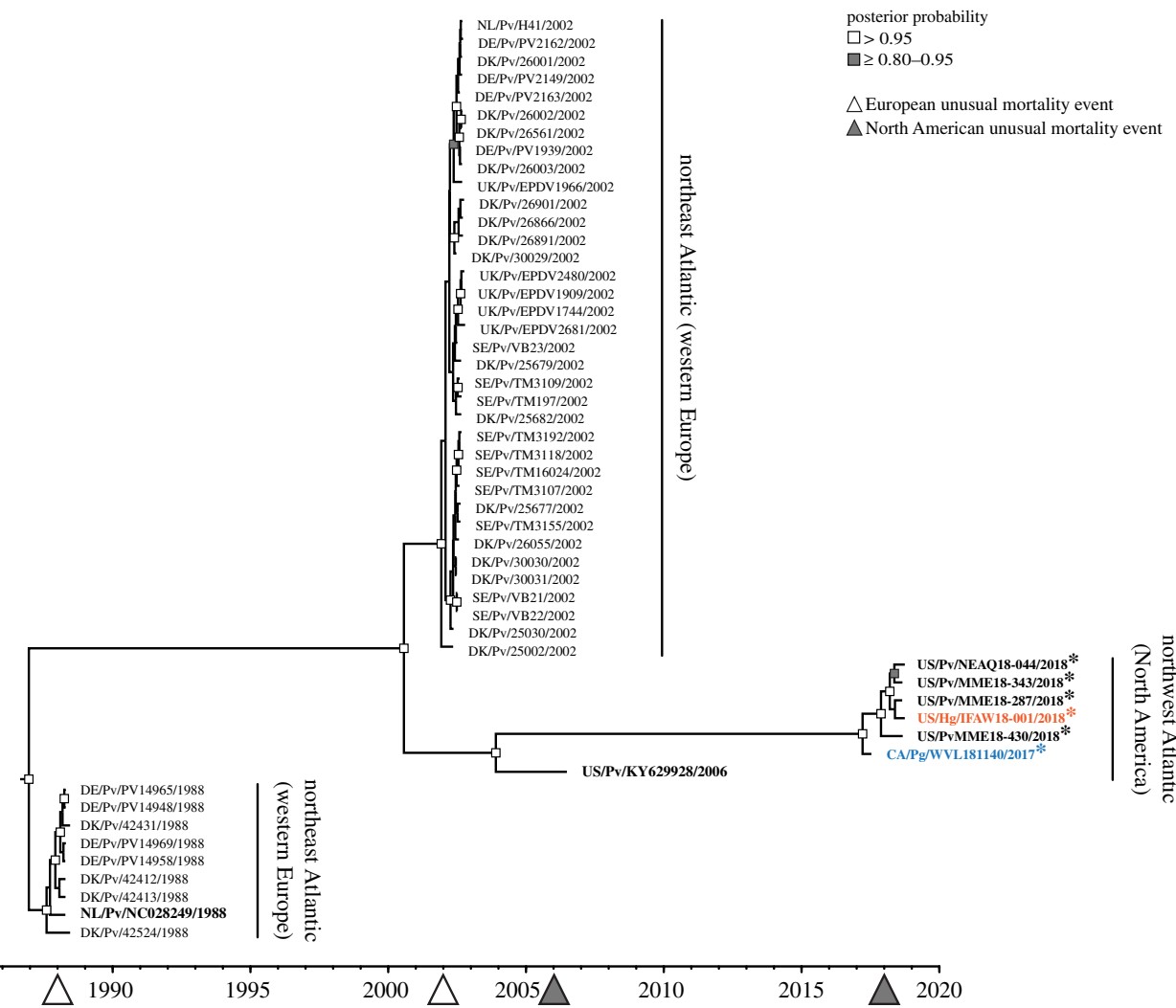

**Figure 2.** Phylogenetic relationships of PDV from the northeast and northwest Atlantic. Newly reported (bold, asterisk) and previously published genomes (bold), and previously published PMFH sequences were analysed to estimate time of most recent common ancestor(s) and lineage. Sequences represent virus derived from harbour (Pv, black), grey (Hg, orange) and harp seal (Pg, blue). Four PDV-related UMEs have occurred in the northeast (white arrows) and northwest (grey arrows) Atlantic. Time-scaled tree was generated with BEAST. Posterior probabilities ≥0.8 and greater than 0.95 are labelled at nodes with white or grey squares, respectively. (Online version in colour.)

a continued presence in the northwest Atlantic. Given that there is an absence of data prior to the first North American UME in 2006, it is currently not possible to determine if PDV circulated endemically within any of the regional pinnipeds prior to the first documented outbreak, or had separate, recurring introductions of highly conserved virus.

Our analysis of archived sera samples from multiple pinniped species support the continued presence of PDV in the northwest Atlantic. We detected seropositive animals in all years where analyses were performed, spanning 2010–2013 (figure 1a). The majority of the data presented here is derived from weanlings. Though there may be some remaining level of maternal antibody, passive immunity declines within a few weeks and is unlikely to explain the relatively sustained average 31.7% seropositivity detected in young animals in the northwest Atlantic several years after a UME [35]. By contrast to the results reported here for the northwest Atlantic, a previously published longitudinal analysis of harbour seals in the northeast Atlantic found that antibodies to PDV steadily decreased in subsequent years after each European UME and were rarely detected at all in pups and juveniles [2]. We also demonstrated the presence of PDV outside of UME years in the northwest Atlantic, detecting viral RNA in each year

studied, spanning 2011 through 2020, and representing stranded animals, wild capture pups, grey, harbour and harp seals (figure 1b,c). We were able to obtain partial to full genome sequence from the majority of years studied.

Differences in species distribution and density may provide a framework from which to consider the apparent ongoing presence of PDV in the northwest Atlantic versus the acute dead-end introductions in the northeast Atlantic. The global harp seal population is in the millions and could readily support endemic infection [4]. Harp seal distributions range across the Arctic but are separated into three populations: northwest Atlantic, West Ice (Greenland) and East Ice (Barents Sea) (figure 5a). Of these, the Northwest Atlantic population is the largest, and is estimated to have stabilized at around 7.6 million animals, while the West and East Ice populations are 300 000 and 1.3 million respectively (figure 5b) [36]. Harp seals are highly migratory, generally following receding pack ice to the high arctic to feed in summer and moving ahead of ice formation southward to breed in winter. While the populations are considered distinct, the West and East Ice populations share significant territory in the north Barents Sea during each summer, and more sporadically, the West Ice population overlaps with

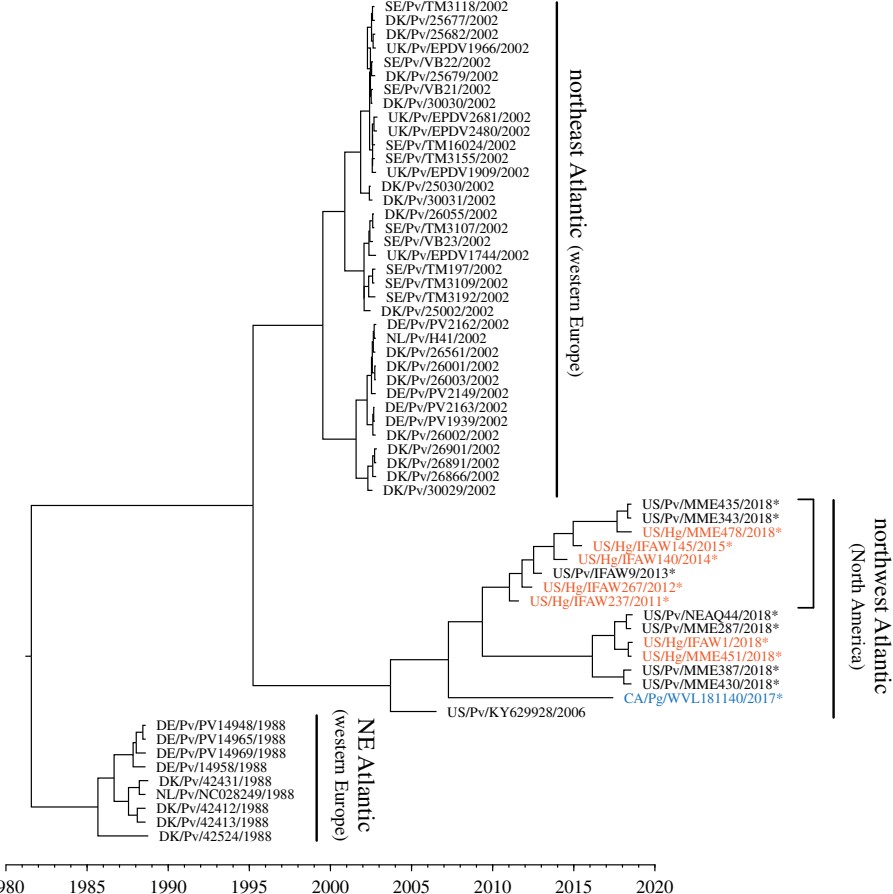

**Figure 3.** PDV H splits into two clusters within the northwest Atlantic. Full-length H sequences were compared using BEAST. Sequences represent viruses from harbour (Pv, black), grey (Hg, orange) and harp (Pg, blue) seals. Dataset comprised 46 previously published sequences and 15 newly reported sequences (asterisk). Sequences with a substitution hotspot from residues 160–236 are bracketed. (Online version in colour.)

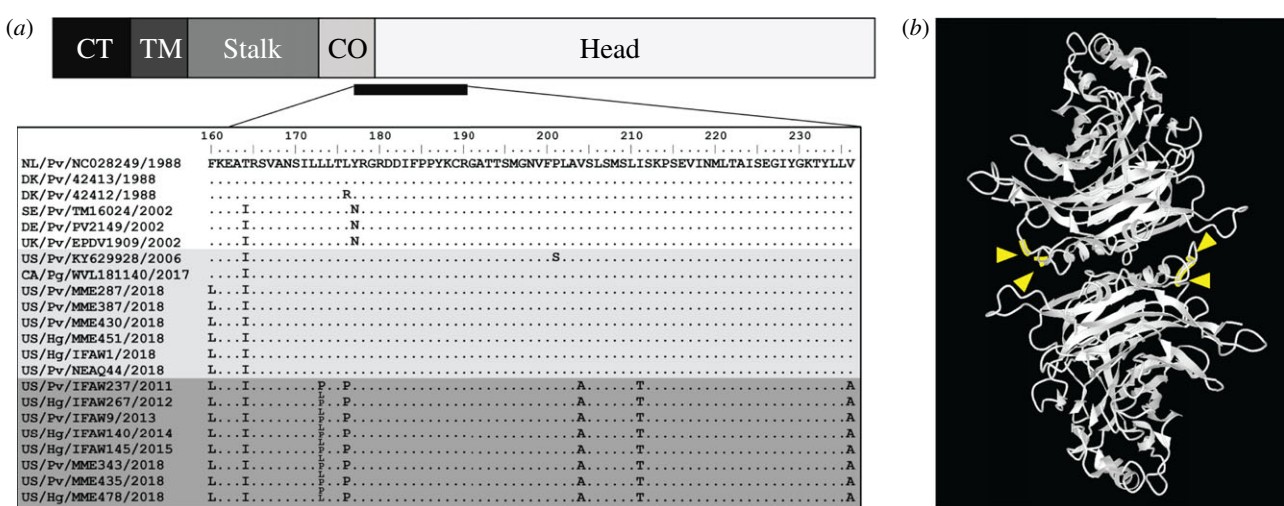

**Figure 4.** Substitution hotspot in endemic PDV in Northwestern Atlantic seals. (*a*) H protein comprises: cytoplasmic tail (CT), transmembrane (TM), stalk, connector (CO) and head. The alignment shown spans the connector and head domain, referenced to NL/Pv/NC028249/1988, with a dot indicating identity. Shading denotes sequence from northeast (white) and northwest Atlantic without (light grey) and with (dark grey) this substitution pattern. The dual L/P at position 172 indicates mixed single-nucleotide variants. (*b*) The H head domain, shown here as a homo-dimer, is the site of 1–2 leucine to proline substitutions (yellow arrows) within the 228 nt substitution hotspot (yellow residues) at the dimer interface. (Online version in colour.)

the northwest Atlantic population during their secondary summer migration routes along the southern Greenland coast [37]. The intermingling of this robust population could support PDV dissemination within harp seals spanning the North Atlantic.

A notable difference in pinniped populations on the western and eastern side of the North Atlantic lies in the frequency of species overlap, particularly between harp seals and other species. In parts of eastern North America, there is a large population of seasonally sympatric harp,

Proc. R. Soc. B 288: 20211841

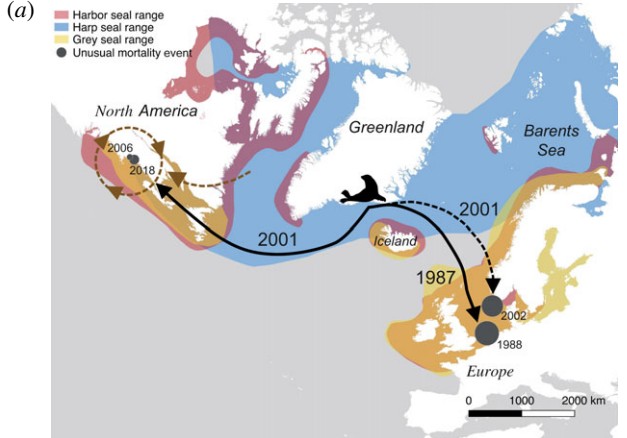

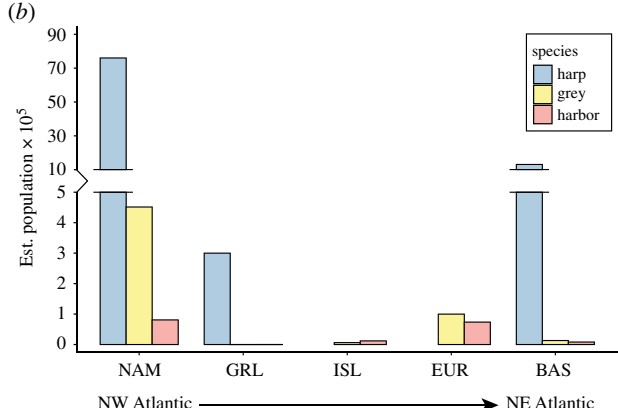

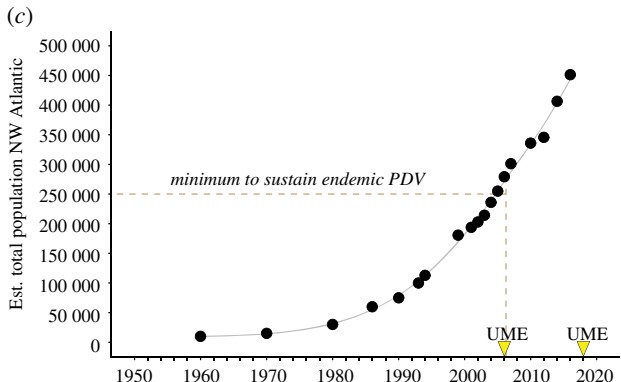

**Figure 5.** Population and species distribution related to PDV outbreaks and circulation. (*a*) North Atlantic distribution of harbour (red), harp (blue) and grey (yellow) seal populations as listed in IUCN Red List. PDV was seeded from an Arctic population into both the northeast and northwest Atlantic (solid black arrows). PDV has had continued circulation in the northwest Atlantic as either endemic virus or repeat seeding (dotted brown lines). Northeast Atlantic was reseeded in 2001 by a second event likely via Arctic populations (dotted black arrow). Molecular dating was used to infer timing of introduction events. The estimated number of mortalities from the four documented UMEs are represented by size-scaled markers (grey circles) and UME year is indicated. (*b*) The estimated population density for harp, grey and harbour seals were binned by geographical region: NAM, North America; GRL, Greenland; ISL, Iceland; EUR, Europe; BAS, Baltic Sea. (*c*) Exponential growth in the estimated total population of northwest Atlantic grey seals passes the estimated minimum population threshold to sustain endemic PDV just prior to the first documented PDV UME in the northwest Atlantic.

grey and harbour seals (figure 5*a*,*b*). The northwest Atlantic harp population shares the Gulf of St Lawrence with a large population of grey seals (estimated 451 431) and

harbour seals (estimated 80 834) [17,36]. Since the late 2000s, the northwest Atlantic grey seal population has repopulated past the estimated minimum level of 250 000 needed to support endemic infection of morbillivirus (figure 5*c*) [17,38–41]. By contrast, West Ice harp seals only overlap with a small, local population of non-migratory Icelandic grey seals (estimated 6269) and the East Ice population of harp seals infrequently overlaps a small population of grey or harbour seal in the Barents Sea (estimated 13 140 and 8142, respectively) [42]. Currently neither the grey nor harbour seal populations in the northeast Atlantic are likely above the estimated minimum of 250 000 animals thought to be necessary to support endemic morbillivirus infection on their own (currently estimated at 120 000 and 54 109, respectively) [42]. This may explain why there is no evidence of ongoing circulation of PDV outside of mortality events shortly following each new introduction to European waters.

Although exposure to PDV from harp seals is possible on both sides of the Atlantic, the extent of habitat overlap and the density of new hosts is likely, to date, only favourable for sustained PDV circulation in other pinnipeds in the northwest but not northeast Atlantic. It would have therefore been possible after the 2006 northwest Atlantic UME to subsequently establish endemic PDV within grey seals in eastern North America.

Through the expansion of PDV sequence made possible through this study, we further genetically characterized an apparently regional PDV variant that emerged in the northwest Atlantic by 2011. This regional variant has a persistent substitution hotspot that we hypothesize will decrease viral fusion and may result in a naturally attenuated virus. This variant was detected in both harbour and grey seals from 2011 through the most recent 2018 UME. We postulate that this variant may be a less virulent strain that causes more moderate illness. This may provide some degree of regional immunity in grey and harbour seals by providing antigen for antibody development with relatively low morbidity and mortality. Though this regional strain appears to have had a sustained presence in grey and harbour seals in the northwest Atlantic in non-UME years, there is currently no evidence that it circulated in harp seals. The single harp seal derived PDV genome recovered in an animal from Canada in 2017 was more similar to those observed in both UMEs in the northwest Atlantic. Thus, we propose that a regional strain of PDV circulates in grey and harbour seals in the northwest Atlantic and that UMEs in the region are triggered by the sporadic introduction of a more virulent PDV strain from outside the region. We further propose that the effect of the resulting UMEs may be limited by previous immunity from the regional PDV strain among sympatric grey and harbour seals, though further study is necessary to investigate these hypotheses.

In conclusion, this study has provided phylogenetic evidence that PDV was introduced into pinnipeds in the northwest Atlantic by a single cross-Atlantic lineage and has now become regionally endemic as local populations have rebounded and ongoing interactions with harp seals have been possible. Serological and viral screening of pinniped surveillance samples from 2006 onward suggest there is continued circulation of PDV outside of UMEs and among multiple pinniped species in the northwest

Atlantic with and without clinical signs. Lastly, we identified a regional variant that we hypothesize may provide some population level protection that limits the severity of outbreaks and could be useful in future modelling efforts evaluating the impact of PDV. High-latitude ecosystems are expected to be increasingly disrupted by climate change, altering the conditions for disease ecology with multiple stressors. The observations reported here help to provide a framework for emerging disease as ecological conditions and species distribution and overlap continue to shift.

Ethics. All grey seal rookery sampling was conducted under National Marine Fisheries Service (NMFS) permits 17670 and 21719. Work on Monomoy Island and Nantucket was performed under National Wildlife Refuge (NWR) Special Use Permits 16-MNY-01 and 53514. Additional seal sampling and transfer of samples were authorized under permit NMSF 21966.

Data accessibility. The datasets supporting this article including sequences, alignments and phylogenetic analyses have been deposited at https://github.com/ksawatzki/Supp_data/.

The data are provided in electronic supplementary material [43].

Authors' contributions. W.P.: conceptualization, data curation, formal analysis, funding acquisition, investigation, methodology, project administration, supervision, writing-original draft, writing-review and editing; K.S.: conceptualization, data curation, formal analysis, funding acquisition, investigation, methodology, visualization, writing-original draft, writing-review and editing; A.B.: conceptualization, investigation, methodology, resources, writing-review and editing; N.H.: formal analysis, methodology, visualization, writing-review and editing; A.F.: formal analysis, investigation, writing-review and editing; I.S., M.TO., O.N., T.W., T.G., K.S., T.C.S.R. and M.B.: data curation, writing-review and editing; L.D.: investigation, resources, writing-review and editing; L.B., A.S., M.N., A.T. and T.R.: resources, writing-review and editing; M.B.L.: investigation, writing-review and editing; D.F.: investigation, project administration, resources, writing-review and editing; J.R.: funding acquisition, resources, supervision, writing-review and editing.

All authors gave final approval for publication and agreed to be held accountable for the work performed therein.

Competing interests. We declare we have no competing interests

Funding. This work was supported by NIAID grant no. HHSN272201400008C/AI/NIAID.

Acknowledgements. The authors thank Judy St. Leger and Keith Matassa for performing serological assays, Pierre-Yves Daoust for contributing the genome CA/Harp/WVL181140/2017, and Brett Smith and Alexandre Tremeau-Bravard for processing samples. We thank Gordon Waring, Kimberly Murray, Elizabeth Josephson, Frederick Wenzel and the rest of the team at National Marine Fisheries Service's Northeast Fisheries Science Center who provide expertise, permitting and logistical support for these studies. We thank Monomoy National Wildlife Refuge, National Park Service, Fish and Wildlife Service, The Trustees of Reservations, and University of Massachusetts Nantucket Field Station for critical assistance in field operations. We are deeply indebted to Marine Mammals of Maine, Atlantic Marine Conservation Society, College of the Atlantic, Seacoast Science Center, National Marine Life Center, International Fund for Animal Welfare, University of New England, University of Connecticut, New England Aquarium, National Aquarium and Mystic Aquarium who obtain samples from stranded and managed care animals and provide resources and personnel for field teams. This work could not have been conducted without the tireless efforts of the Greater Atlantic Region Marine Mammal Stranding Network in responding to and collecting samples from stranded seals. We thank Mendy Garron and Ainsley Smith from the National Marine Fisheries Service's Greater Atlantic Region Marine Mammal Stranding Response Program for managing availability of samples from stranded animals and operations during the UME. We also thank the members of the Working Group for the Marine Mammal Unusual Mortality Events and the members of the UME Investigative Teams. We thank Robert DiGiovanni, Dominique Walk, Katie Gilbert and Megan Ely for field assistance during the 2018 UME. Finally, we are forever grateful to the many members of the field teams who have made it possible to obtain samples from the rookeries. This constitutes scientific contribution #321 from the Sea Research Foundation, Inc., d/b/a Mystic Aquarium. The scientific results and conclusions, as well as any views or opinions expressed herein, are those of the authors and do not necessarily reflect the views of NOAA.

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
