## [Peer Review File · Proceedings of the Royal Society B: Biological Sciences]

Review History

RSPB-2021-0437.R0 (Original submission)

Review form: Reviewer 1

Recommendation

Accept with minor revision (please list in comments)

Scientific importance: Is the manuscript an original and important contribution to its field?

Good

General interest: Is the paper of sufficient general interest?

Good

Quality of the paper: Is the overall quality of the paper suitable?

Good

Is the length of the paper justified?

Yes

Should the paper be seen by a specialist statistical reviewer?

No

Do you have any concerns about statistical analyses in this paper? If so, please specify them explicitly in your report.

Yes

It is a condition of publication that authors make their supporting data, code and materials available - either as supplementary material or hosted in an external repository. Please rate, if applicable, the supporting data on the following criteria.

Is it accessible?

No

Is it clear?

No

Is it adequate?

No

Do you have any ethical concerns with this paper?

No

Comments to the Author

Phocine distemper virus is an ecologically complex disease system and the authors do a commendable job of trying to piece together how multiple seal species located in North America, Greenland, and Europe intersect to cause periodic outbreaks of PDV. The genomic data adds an important dimension to the story that leads to an intriguing hypothesis that mutations led to an attenuated strain that became endemic in the North America population and this explains high rates of seropositivity but less severe outbreaks in American seals. But it's a complex story with a lot of missing data and at times it can be difficult to piece it together the way it's presented.

1. Can you be clearer in the Methods/Results/Legend what sequences are included in Figure 1? What is the length (in nucleotides) of the full-length genomes versus Phosphoprotein-Matrix-Fusion-Hemagglutinin gene? If there is a large difference in sequence length it would make sense to also infer a tree just for the P-M-F-H as comparison.
2. Figure 5 contains a number of over-interpretations. It is impossible to infer from Figures 1 and 3 that endemic circulation of PDV began in the Northwest Atlantic population in 1987. The 2006 virus and 2017-2018 viruses share a common ancestor in the 2000s, as far back as you can reasonably infer, not to the TMRCA of the entire tree. And that is when the seal population reached the minimum population size needed to sustain endemic transmission. The dotted arrow with the 2001 trans-Atlantic movement is therefore highly speculative and should be removed.
3. The rooting is confusing in Figure 3. For consistency could you root the tree similar to Figure 1, rooted by the oldest 1988 viruses? Otherwise it's difficult to reconcile the two trees, particularly the US 2006 virus.
4. Is there any experimental evidence that the US viruses are attenuated?

Minor

1. Bootstrap values indicate support for specific nodes, not branches, and should be placed

next to nodes in Figure 3.

2. What's going on with Tiger and Cougar in Figure S3? Is that a misalignment? Sequencing errors?
3. It would be helpful to label the virus strain names more simply in the phylogenetic trees (remove the month and date at the end of the name so it's just year).
4. The map of the seal species ranges in Figure 5 is nice. Is it possible to include a map that shows the spatial differences in animal density? For example to show how European numbers are smaller? Can you describe a little more about what is known about long-distance movements/ranges of these seal species? Some more background on the ecology would be helpful.

Review form: Reviewer 2

Recommendation

Major revision is needed (please make suggestions in comments)

Scientific importance: Is the manuscript an original and important contribution to its field?

Good

General interest: Is the paper of sufficient general interest?

Excellent

Quality of the paper: Is the overall quality of the paper suitable?

Good

Is the length of the paper justified?

Yes

Should the paper be seen by a specialist statistical reviewer?

No

Do you have any concerns about statistical analyses in this paper? If so, please specify them explicitly in your report.

Yes

It is a condition of publication that authors make their supporting data, code and materials available - either as supplementary material or hosted in an external repository. Please rate, if applicable, the supporting data on the following criteria.

Is it accessible?

No

Is it clear?

N/A

Is it adequate?

N/A

Do you have any ethical concerns with this paper?

No

Comments to the Author

This paper examines the phenomenon of phocine distemper virus outbreaks in Atlantic pinniped populations, which have caused significant mortality in some cases and received a lot of scientific and public interest over the past decades. One of the main conundrums about these outbreaks is why they appear to be limited to the eastern Atlantic, whereas outbreaks of comparable severity have not been seen in populations of the same species on the western side of the Atlantic. The study seeks to provide new answers to this question through a combination of viral genetic and seal host serology data. Based on these data, the author argue that continuous circulation of a less pathogenic form of the virus in the northwest Atlantic might maintain high enough levels of immunity with these populations to prevent the kind of outbreaks seen in eastern populations.

I found the study interesting and well presented but I am not convinced by the authors' data and conclusions. Specifically, the argument about endemic circulation creating partial herd immunity rests on such endemic viruses being limited to western Atlantic populations but being absent in the East. Published data to document this might exist but if so they should be presented for comparison. The part about genetic differentiation of the viruses on either side of the Atlantic is interesting but doesn't appear to be novel. The last part of the study, focussed on variation in the virus' hemagglutinin gene which the authors argue is responsible for different virus phenotypes, seems very speculative and likely involves incorrect analyses (see below). I appreciate that the data for this type of work are hard to come by and that the current study is exceptional in terms of the amount of data that has been assembled. Still, the study would benefit from strengthening the arguments where possible but to otherwise refrain from speculative claims and present the findings as hypothesis-generating rather than confirmatory. There are also a several methodological issues that need to be addressed.

Specific points:

1. Based on serological and qPCR results, the authors argue that some form of PDV is circulating endemically in North American seals. For comparison, do we know that such evidence of consistent exposure is definitely not detectable in European populations? The authors cite data from the eastern Atlantic comparable to their own in the discussion but for the readers to evaluate this comparison it would be helpful to include these published data in their figures or tables. It is also not clear to me whether the available data are limited to serology or whether the same patterns hold when screening eastern populations by qPCR for circulating PDV.
2. Please indicate which sequences in Fig 1 were new and which had been previously published and analysis. This would make it easier to appreciate what the current study is able to add to previous work and what had been known before in terms of geographical clades and their divergence times. It would also be helpful to include highest posterior density intervals on the key internal nodes (i.e. split between NE and NW Atlantic).
3. Before attempting to reconstruct time-scaled phylogenies it would be important to confirm that there is actually enough temporary signal (increase in divergence over time) to estimate a molecular clock. This should be done based on the ML phylogeny using TempEst and results could go into the supplement rather than the main text.
4. Results of the model selection for molecular clock and demographic priors should be documented - the authors state that they decided on an exponential growth model but we don't know what other models were considered. Exponential growth might not be an obvious choice for a virus that appears to be circulating endemically and the choice of demographic prior can have a significant effect on estimated divergence times.
5. The selection analysis in Table S4 looks wrong: the simpler models (M1a, M7) have a much higher likelihood than their complex counterparts with two extra parameters, which shouldn't be

the case (additional parameters might fail to improve the likelihood but they can't reduce it). This suggests that some of the PAML analyses didn't converge and that the results are unreliable. Given that there appears to be a lack of sites experiencing more than one non-synonymous substitution, I would be surprised if the data contained any evidence of positive selection. I don't think it is appropriate to use the BEAST tree as the topology for the PAML analysis, this should be a tree estimated without a molecular clock.

6. L378-382. I don't understand the point made in this paragraph - genetic variation was found in this position, but how is this evidence of intra-host selection?

7. L391-393. The effect of these mutations on the virus ability for fusion is complete speculation - it is not appropriate to refer to this as a 'low-fusion lineage' in the Discussion

minor comments:

- description of virus genomic sequencing is not consistent (e.g. read length, paired/unpaired reads). Please add missing information
- line 430 - I believe this should say "Northeast Atlantic"?
- the scale bars on the phylogenetic trees represent the 'substitution rate' not 'mutation rate' - please change throughout
- I like Fig. 5 but I find it very difficult to read, especially the part under the time line in the right panel
- Highlight species of interest in the trees shown in supplemental figures S2 and S3?
- how was the tree in Fig 3 rooted?
- the trees would benefit from better formatting: if the viewer is supposed to evaluate the position of specific taxa (e.g. 'orphan sequences', sequences from UME or non-UME events, different host species), these should be visually highlighted in the tree

Decision letter (RSPB-2021-0437.R0)

19-Apr-2021

Dear Dr Sawatzki:

I am writing to inform you that your manuscript RSPB-2021-0437 entitled "Longitudinal analysis of pinnipeds in the Northwest Atlantic provides insights on endemic circulation and regional immunity to Phocine distemper virus" has, in its current form, been rejected for publication in Proceedings B.

This action has been taken on the advice of referees, who have recommended that substantial revisions are necessary. With this in mind we would be happy to consider a resubmission, provided the comments of the referees are fully addressed. However please note that this is not a provisional acceptance.

Sincerely,
 Professor Hans Heesterbeek
 mailto: proceedingsb@royalsociety.org

Associate Editor
 Board Member: 1
 Comments to Author:

Both reviewers recognize the potential contribution of this work but raise substantial concerns that must be addressed, both regarding methodological approaches and presentation of results.

Reviewer(s)' Comments to Author:

Referee: 1

Comments to the Author(s)

Phocine distemper virus is an ecologically complex disease system and the authors do a commendable job of trying to piece together how multiple seal species located in North America, Greenland, and Europe intersect to cause periodic outbreaks of PDV. The genomic data adds an important dimension to the story that leads to an intriguing hypothesis that mutations led to an attenuated strain that became endemic in the North America population and this explains high rates of seropositivity but less severe outbreaks in American seals. But it's a complex story with a lot of missing data and at times it can be difficult to piece it together the way it's presented.

1. Can you be clearer in the Methods/Results/Legend what sequences are included in Figure 1? What is the length (in nucleotides) of the full-length genomes versus Phosphoprotein-Matrix-Fusion-Hemagglutinin gene? If there is a large difference in sequence length it would make sense to also infer a tree just for the P-M-F-H as comparison.

2. Figure 5 contains a number of over-interpretations. It is impossible to infer from Figures 1 and 3 that endemic circulation of PDV began in the Northwest Atlantic population in 1987. The 2006 virus and 2017-2018 viruses share a common ancestor in the 2000s, as far back as you can reasonably infer, not to the TMRCA of the entire tree. And that is when the seal population reached the minimum population size needed to sustain endemic transmission. The dotted arrow with the 2001 trans-Atlantic movement is therefore highly speculative and should be removed.

3. The rooting is confusing in Figure 3. For consistency could you root the tree similar to Figure 1, rooted by the oldest 1988 viruses? Otherwise it's difficult to reconcile the two trees, particularly the US 2006 virus.

4. Is there any experimental evidence that the US viruses are attenuated?

Minor

1. Bootstrap values indicate support for specific nodes, not branches, and should be placed next to nodes in Figure 3.

2. What's going on with Tiger and Cougar in Figure S3? Is that a misalignment? Sequencing errors?

3. It would be helpful to label the virus strain names more simply in the phylogenetic trees (remove the month and date at the end of the name so it's just year).

4. The map of the seal species ranges in Figure 5 is nice. Is it possible to include a map that shows the spatial differences in animal density? For example to show how European numbers are smaller? Can you describe a little more about what is known about long-distance movements/ranges of these seal species? Some more background on the ecology would be helpful.

Referee: 2

Comments to the Author(s)

This paper examines the phenomenon of phocine distemper virus outbreaks in Atlantic pinniped populations, which have caused significant mortality in some cases and received a lot of scientific and public interest over the past decades. One of the main conundrums about these outbreaks is why they appear to be limited to the eastern Atlantic, whereas outbreaks of comparable severity have not been seen in populations of the same species on the western side of the Atlantic. The study seeks to provide new answers to this question through a combination of viral genetic and seal host serology data. Based on these data, the author argue that continuous circulation of a less pathogenic form of the virus in the northwest Atlantic might maintain high enough levels of immunity with these populations to prevent the kind of outbreaks seen in eastern populations.

I found the study interesting and well presented but I am not convinced by the authors' data and conclusions. Specifically, the argument about endemic circulation creating partial herd immunity rests on such endemic viruses being limited to western Atlantic populations but being absent in the East. Published data to document this might exist but if so they should be presented for comparison. The part about genetic differentiation of the viruses on either side of the Atlantic is interesting but doesn't appear to be novel. The last part of the study, focussed on variation in the virus' hemagglutinin gene which the authors argue is responsible for different virus phenotypes, seems very speculative and likely involves incorrect analyses (see below). I appreciate that the data for this type of work are hard to come by and that the current study is exceptional in terms of the amount of data that has been assembled. Still, the study would benefit from strengthening the arguments where possible but to otherwise refrain from speculative claims and present the findings as hypothesis-generating rather than confirmatory. There are also a several methodological issues that need to be addressed.

Specific points:

1. Based on serological and qPCR results, the authors argue that some form of PDV is circulating endemically in North American seals. For comparison, do we know that such evidence of consistent exposure is definitely not detectable in European populations? The authors cite data from the eastern Atlantic comparable to their own in the discussion but for the readers to evaluate this comparison it would be helpful to include these published data in their figures or tables. It is also not clear to me whether the available data are limited to serology or whether the same patterns hold when screening eastern populations by qPCR for circulating PDV.

2. Please indicate which sequences in Fig 1 were new and which had been previously published and analysis. This would make it easier to appreciate what the current study is able to add to previous work and what had been known before in terms of geographical clades and their divergence times. It would also be helpful to include highest posterior density intervals on the key internal nodes (i.e. split between NE and NW Atlantic).

3. Before attempting to reconstruct time-scaled phylogenies it would be important to confirm that there is actually enough temporary signal (increase in divergence over time) to estimate a molecular clock. This should be done based on the ML phylogeny using TempEst and results could go into the supplement rather than the main text.

4. Results of the model selection for molecular clock and demographic priors should be documented - the authors state that they decided on an exponential growth model but we don't know what other models were considered. Exponential growth might not be an obvious choice for a virus that appears to be circulating endemically and the choice of demographic prior can have a significant effect on estimated divergence times.

5. The selection analysis in Table S4 looks wrong: the simpler models (M1a, M7) have a much higher likelihood than their complex counterparts with two extra parameters, which shouldn't be the case (additional parameters might fail to improve the likelihood but they can't reduce it). This suggests that some of the PAML analyses didn't converge and that the results are unreliable. Given that there appears to be a lack of sites experiencing more than one non-synonymous substitution, I would be surprised if the data contained any evidence of positive selection. I don't think it is appropriate to use the BEAST tree as the topology for the PAML analysis, this should be a tree estimated without a molecular clock.

6. L378-382. I don't understand the point made in this paragraph - genetic variation was found in this position, but how is this evidence of intra-host selection?

7. L391-393. The effect of these mutations on the virus ability for fusion is complete speculation - it is not appropriate to refer to this as a 'low-fusion lineage' in the Discussion

minor comments:

- description of virus genomic sequencing is not consistent (e.g. read length, paired/unpaired reads). Please add missing information
- line 430 - I believe this should say "Northeast Atlantic"?
- the scale bars on the phylogenetic trees represent the 'substitution rate' not 'mutation rate' - please change throughout
- I like Fig. 5 but I find it very difficult to read, especially the part under the time line in the right panel
- Highlight species of interest in the trees shown in supplemental figures S2 and S3?
- how was the tree in Fig 3 rooted?
- the trees would benefit from better formatting: if the viewer is supposed to evaluate the position of specific taxa (e.g. 'orphan sequences', sequences from UME or non-UME events, different host species), these should be visually highlighted in the tree

Author's Response to Decision Letter for (RSPB-2021-0437.R0)

See Appendix A.

RSPB-2021-1841.R0

Review form: Reviewer 2

Recommendation

Accept with minor revision (please list in comments)

Scientific importance: Is the manuscript an original and important contribution to its field?

Excellent

General interest: Is the paper of sufficient general interest?

Good

Quality of the paper: Is the overall quality of the paper suitable?

Good

Is the length of the paper justified?

Yes

Should the paper be seen by a specialist statistical reviewer?

No

Do you have any concerns about statistical analyses in this paper? If so, please specify them explicitly in your report.

No

It is a condition of publication that authors make their supporting data, code and materials available - either as supplementary material or hosted in an external repository. Please rate, if applicable, the supporting data on the following criteria.

Is it accessible?

Yes

Is it clear?

Yes

Is it adequate?

Yes

Do you have any ethical concerns with this paper?

No

Comments to the Author

The authors did a commendable job incorporating the reviewers' comments and have made it much more clear what the contribution of their study is and what their key findings are. I only have a few specific points, listed below, which I suggest the authors should look.

One general point is that the authors could do more to convey the broader importance of their work. The abstract for example is quite focussed on the virus studied but provides little indication as to why we need to know these things and what can be taken away from the study. Instead of reporting in the final sentence what sequence data have been generated, I would expect to see some general conclusions, possibly beyond PDV. Similar opportunities exist in the Introduction and the Discussion.

specific points:

I appreciate the additional information provided about the marginal likelihood estimation and model comparison in BEAST in the supplement. Were those done using default settings? If so, please state that or otherwise provide further detail. Some of the results look questionable - the two methods (stepping stone and path sampling) should produce very similar estimates but for some of the models, including for the model selected as the top one, there are major differences, in the order of several hundred log units. This suggests that these analyses were probably not run for a sufficient length of time (steps). I don't expect this change the main conclusions of the paper but it does not look like good practice.

It is not clear how the authors arrived at the "estimated minimum level of 250,000 needed to support endemic infection of morbillivirus" (line 1108-1110). This is a really interesting point (mirroring what is seen in other morbilliviruses like measles, maybe cite some of this work?). The cited reports look like they would contain estimated seal numbers but did they also report this persistence threshold for PDV? If so, how was it inferred?

line 545 - poorly constructed sentence: "Individual PDV genes were aligned for... genomes..." - why not simply "Sequences were aligned using Clustal"?

line 550 - how were dates formatted when only year was available? Or where all dates simplified to 'year only'?

line 554 - should say "RELAXED CLOCK with an uncorrelated lognormal distribution"

line 798 - I wonder about the heading: to me the key finding here is not the single introduction (there might have been others that simply weren't detected) but the continuous circulation of a single PDV lineage in the Northeastern Atlantic since the early 2000's

line 816, 997, 1128 - 'predicted' should be replaced with 'hypothesised'. The word prediction has a specific meaning in science and generally involves some quantitative (i.e. statistical) basis so it doesn't seem appropriate here.

line 825 - "...and is referred to ..." should be "... with the latter being referred to..."

Fig 3 would benefit from some additional formatting to help the reader. For example, it would be helpful to distinguish the Northwestern and Northeastern sequences as in Fig. 2. The colour used on indicate harp seals looks different from the one in Fig 2.

Decision letter (RSPB-2021-1841.R0)

23-Sep-2021

Dear Dr Sawatzki:

Your manuscript has now been peer reviewed and the reviews have been assessed by an Associate Editor. The reviewer's comments (not including confidential comments to the Editor) and the comments from the Associate Editor are included at the end of this email for your reference. As you will see, the reviewer and the Associate Editor have raised some issues and we would like to invite you to revise your manuscript to address them.

We do not allow multiple rounds of revision so we urge you to make every effort to fully address all of the comments at this stage. If deemed necessary by the Associate Editor, your manuscript

will be sent back to one or more of the original reviewers for assessment. If the original reviewers are not available we may invite new reviewers. Please note that we cannot guarantee eventual acceptance of your manuscript at this stage.

Research ethics:

Use of animals and field studies:

It is a condition of publication that you make available the data and research materials supporting the results in the article (<https://royalsociety.org/journals/authors/author-guidelines/#data>). Datasets should be deposited in an appropriate publicly available repository and details of the associated accession number, link or DOI to the datasets must be included in the Data Accessibility section of the article (<https://royalsociety.org/journals/ethics-policies/data-sharing-mining/>). Reference(s) to datasets should also be included in the reference list of the article with DOIs (where available).

Please submit a copy of your revised paper within three weeks. If we do not hear from you within this time your manuscript will be rejected. If you are unable to meet this deadline please let us know as soon as possible, as we may be able to grant a short extension.

Best wishes,
Professor Hans Heesterbeek
mailto:proceedingsb@royalsociety.org

Associate Editor Board Member

Comments to Author:

Thank you for addressing the reviewers' comments in this revision. We have sent the manuscript back to one of the original referees, who has provided additional suggestions. In addition to addressing these critiques, please consider (a) moving some of the additional methodological detail added in this version on generation of the genomes into the supplement and (b) providing access to the serological data (in addition to the genetic data, which we note is already on GenBank).

Reviewer(s)' Comments to Author:

Referee: 2

Comments to the Author(s).

The authors did a commendable job incorporating the reviewers' comments and have made it much more clear what the contribution of their study is and what their key findings are. I only have a few specific points, listed below, which I suggest the authors should look.

One general point is that the authors could do more to convey the broader importance of their work. The abstract for example is quite focussed on the virus studied but provides little indication as to why we need to know these things and what can be taken away from the study. Instead of reporting in the final sentence what sequence data have been generated, I would expect to see some general conclusions, possibly beyond PDV. Similar opportunities exist in the Introduction and the Discussion.

specific points:

I appreciate the additional information provided about the marginal likelihood estimation and model comparison in BEAST in the supplement. Were those done using default settings? If so, please state that or otherwise provide further detail. Some of the results look questionable - the two methods (stepping stone and path sampling) should produce very similar estimates but for some of the models, including for the model selected as the top one, there are major differences,

in the order of several hundred log units. This suggests that these analyses were probably not run for a sufficient length of time (steps). I don't expect this change the main conclusions of the paper but it does not look like good practice.

It is not clear how the authors arrived at the "estimated minimum level of 250,000 needed to support endemic infection of morbillivirus" (line 1108-1110). This is a really interesting point (mirroring what is seen in other morbilliviruses like measles, maybe cite some of this work?). The cited reports look like they would contain estimated seal numbers but did they also report this persistence threshold for PDV? If so, how was it inferred?

line 545 - poorly constructed sentence: "Individual PDV genes were aligned for... genomes..." - why not simply "Sequences were aligned using Clustal"?

line 550 - how were dates formatted when only year was available? Or where all dates simplified to 'year only'?

line 554 - should say "RELAXED CLOCK with an uncorrelated lognormal distribution"

line 798 - I wonder about the heading: to me the key finding here is not the single introduction (there might have been others that simply weren't detected) but the continuous circulation of a single PDV lineage in the Northeastern Atlantic since the early 2000's

line 816, 997, 1128 - 'predicted' should be replaced with 'hypothesised'. The word prediction has a specific meaning in science and generally involves some quantitative (i.e. statistical) basis so it doesn't seem appropriate here.

line 825 - "...and is referred to ..." should be "... with the latter being referred to..."

Fig 3 would benefit from some additional formatting to help the reader. For example, it would be helpful to distinguish the Northwestern and Northeastern sequences as in Fig. 2. The colour used on indicate harp seals looks different from the one in Fig 2.

Author's Response to Decision Letter for (RSPB-2021-1841.R0)

See Appendix B.

Decision letter (RSPB-2021-1841.R1)

18-Oct-2021

Dear Dr Sawatzki

I am pleased to inform you that your manuscript RSPB-2021-1841.R1 entitled "Longitudinal analysis of pinnipeds in the Northwest Atlantic provides insights on endemic circulation of Phocine distemper virus" has been accepted for publication in Proceedings B.

The Associate editor has recommended publication, but also suggests some minor revisions to your manuscript. Therefore, I invite you to respond to the comment and revise your manuscript. Because the schedule for publication is very tight, it is a condition of publication that you submit

the revised version of your manuscript within 7 days. If you do not think you will be able to meet this date please let us know.

[http://datadryad.org/submit?journalID=RSPB&manu=\(Document not available\)](http://datadryad.org/submit?journalID=RSPB&manu=(Document%20not%20available)) which will take you to your unique entry in the Dryad repository. If you have already submitted your data to dryad you can make any necessary revisions to your dataset by following the above link. Please see <https://royalsociety.org/journals/ethics-policies/data-sharing-mining/> for more details.

Sincerely,
Professor Hans Heesterbeek
Editor, Proceedings B
<mailto:proceedingsb@royalsociety.org>

Associate Editor:

Comments to Author:

Thank you for your thorough response to the reviewer comments. I also appreciate that you've now made the additional data available on GitHub but would appreciate if additional documentation could be added to the README file detailing what the files are and what's in them.

Author's Response to Decision Letter for (RSPB-2021-1841.R1)

See Appendix C.

Decision letter (RSPB-2021-1841.R2)

19-Oct-2021

Dear Dr Sawatzki

I am pleased to inform you that your manuscript entitled "Longitudinal analysis of pinnipeds in the Northwest Atlantic provides insights on endemic circulation of Phocine distemper virus" has been accepted for publication in Proceedings B.

Data Accessibility section

Open Access

Paper charges

Sincerely,

Appendix A

Attn: Editorial Board, *Proceedings of the Royal Society B*, August 18, 2021

Re: Cover letter for submission of research article

We are submitting the revised manuscript **Longitudinal analysis of pinnipeds in the Northwest Atlantic provides insights on endemic circulation of Phocine distemper virus** for consideration as a research article in *Proceedings of the Royal Society B*.

Our manuscript addresses a question that has perplexed marine wildlife ecologists and virologists for decades: Why does Phocine distemper virus cause massive mortality events in Northern European waters, but not along the North American Atlantic coast?

We appreciate the Editors interest in giving our manuscript further consideration. We are grateful to the reviewers for their careful reading of our work and the insightful suggestions on data analyses and interpretation. We are pleased that both reviewers were intrigued by the data and saw how it contributed to our knowledge of PDV. We also appreciate feedback on places where the narrative was unclear and for the suggestions on ways to clarify the data presentation.

We have incorporated all suggestions from both reviewers and believe that we now present a much more robust and clearly presented representation of the data. The three overarching critiques involved 1) bioinformatic methodology concerns, which we addressed and clarified, 2) interpretations of the more speculative pieces of data, which we either removed all together or softened the interpretations, and 3) complex flow of the data components, which we have reformatted and streamlined for clarity to readers.

Full responses and revisions are found at the end of this letter.

We believe that the work presented in this manuscript is an interesting story and important contribution to viral ecology. We believe it will be of interest to readers from multiple disciplines including virology, epidemiology, ecology and animal sciences. Thank you for your continued consideration of this manuscript for publication in *Proceedings of the Royal Society B*.

Sincerely,

Kaitlin M. Sawatzki and Wendy B. Puryear, on behalf of all co-authors

Reviewer(s)' Comments to Author:

Reviewer comments are in black

Author responses are in blue

Referee: 1

Comments to the Author(s)

Phocine distemper virus is an ecologically complex disease system and the authors do a commendable job of trying to piece together how multiple seal species located in North America, Greenland, and Europe intersect to cause periodic outbreaks of PDV. The genomic data adds an important dimension to the story that leads to an intriguing hypothesis that mutations led to an attenuated strain that became endemic in the North America population and this explains high rates of seropositivity but less severe outbreaks in American seals. But it's a complex story with a lot of missing data and at times it can be difficult to piece it together the way it's presented.

We agree on the complexity of the story and the difficulty in how best to present the interwoven pieces. For this resubmission, we did a substantial rework on the flow of the manuscript and how the data is presented and cross-referenced, we unified how the data was referred across the figures and throughout the text. We removed the more speculative and tangential components of the manuscript. As such, the tables and figures are now in a different order than they were in the initial submission and some have been removed.

The flow of the data presentation that remains in the new format is:

- There is evidence of ongoing PDV in North America outside of UME events, and PDV is present in different species, ages, and clinical presentations (data from archived samples, strandings, and live-captures)
- Genetic sequence from both virus and host support the observed absence of species barrier to PDV infection that we observe in North America
- Phylogeny from new sequences support a single incursion into North America
- Within North America, we found evidence for the ongoing presence of PDV in the population and a regional lineage, in addition to the UME associated lineage

Additional changes that were made to improve clarity:

- We relabeled sequence and animal IDs to standardize across all tables and figures.
- We added a standardized color scheme across figures
- We unified how regions are referred to and wherever possible, standardized the text to be Northwest Atlantic and Northeast Atlantic
- We removed the discussion on the possible reseeding of PDV into Europe from virus circulating in North America and adjusted our interpretation of the incursion timeframe for North America.

- We softened the interpretations related to a putative impact on fusogenicity in the regional strain identified in this study and removed the PAML analysis looking at possible positive selection.

We believe that the new format is significantly more streamlined to provide better clarity on how the most important pieces fit together and that the remaining interpretations are well supported by the presented data.

1. Can you be clearer in the Methods/Results/Legend what sequences are included in Figure 1? What is the length (in nucleotides) of the full-length genomes versus Phosphoprotein-Matrix-Fusion-Hemagglutinin gene? If there is a large difference in sequence length it would make sense to also infer a tree just for the P-M-F-H as comparison.

The figures have been reordered and the previous Figure 1 is now Figure 2. The sequences that were used for all figures and tables have been better described and annotated throughout the text, figures, and legends. Whenever newly derived sequences are shown in tables or Figures, they are marked with an asterisk (currently Figure 2, 3, 4; Supplemental Table S2, S3). Wherever a sequence dataset is utilized (phylogenies of genome and H gene) the precise composition of the included sequences and whether or not they are newly reported here or previously published, is described in both the methods and results.

The sequence length for the full genome is 15,696 nucleotides and the partial genome of P-M-F-H is 7,268. The full analysis was run on the composite sample set reported here (8 full available genomes, combined with 44 P-M-F-H sequences from prior publications) in order to preserve all available data within the model. This was selected to preserve resolution of the North American clade, particularly given the relatively small number of available sequences. An additional tree was generated with all sequences truncated to the P-M-F-H region and the resulting phylogeny was highly comparable to that from the tree presented in the manuscript with only minor changes in highly similar tips. The additional tree with truncated sequence and all related output files are now available on the github repository (https://github.com/ksawatzki/Supp_data/), and text and a supplemental figure added (lines 297-298, Figure S5).

2. Figure 5 contains a number of over-interpretations. It is impossible to infer from Figures 1 and 3 that endemic circulation of PDV began in the Northwest Atlantic population in 1987. The 2006 virus and 2017-2018 viruses share a common ancestor in the 2000s, as far back as you can reasonably infer, not to the TMRCA of the entire tree. And that is when the seal population reached the minimum population size needed to sustain endemic transmission. The dotted arrow with the 2001 trans-Atlantic movement is therefore highly speculative and should be removed.

We appreciate the reviewers suggestions on how to improve the summary map and related text for Figure 5 and have revised accordingly. The goal for Figure 5 is to summarize what is well supported with what we hypothesize based on the new data presented here, so we have tried to more clearly make that distinction in this revision. We removed all discussion of the possible seeding of Europe in 2001 from North America and the trans-Atlantic dotted arrow has been removed. We have also adjusted the estimated incursion to state “by 2001” and referenced that

a more precise estimate is currently hindered by limited sequence availability, particularly from North America prior to 2000.

3. The rooting is confusing in Figure 3. For consistency could you root the tree similar to Figure 1, rooted by the oldest 1988 viruses? Otherwise it's difficult to reconcile the two trees, particularly the US 2006 virus.

We replaced the H gene tree with the original paired supplemental figure (BEAST) to match the type of analysis performed in Figure 2. All viral phylogenetic trees are now time-scaled.

4. Is there any experimental evidence that the US viruses are attenuated?

No. We do not yet have experimental evidence that the regional North American variant is attenuated. Our analysis is currently *in silico* but provides a mechanistically grounded hypothesis. We have removed several points of discussion on this variant and softened the remaining language throughout the manuscript to reflect the fact that the phenotype of this variant is currently speculative.

Minor

1. Bootstrap values indicate support for specific nodes, not branches, and should be placed next to nodes in Figure 3.

This tree has been replaced so this no longer applies.

2. What's going on with Tiger and Cougar in Figure S3? Is that a misalignment? Sequencing errors?

This now refers to supplemental Figure S4. The tiger and cougar Nectin-4 sequences do cluster with the other large cats, but have significantly longer branch lengths than any of the other sequences. The alignment is robust and there are no reported issues with the available sequence that was used in this analysis, though Nectin-4 sequence from both tiger and cougar each contain multiple insertions and deletions. It is unclear to us as to why those particular species are so unusual, but they are from the RefSeq annotations for both animals.

3. It would be helpful to label the virus strain names more simply in the phylogenetic trees (remove the month and date at the end of the name so it's just year).

This change has been made throughout the manuscript text, figures, and tables. Where possible, we also standardized the naming scheme of the sequences to reflect country/species/identifier/year.

4. The map of the seal species ranges in Figure 5 is nice. Is it possible to include a map that shows the spatial differences in animal density? For example to show how European numbers are smaller? Can you describe a little more about what is known about long-distance

movements/ranges of these seal species? Some more background on the ecology would be helpful.

Two paragraphs have been added to the discussion to describe population numbers and movement of all 3 species and on both sides of the Atlantic (lines 364-394). We agree that a map that includes animal density would be a helpful tool and based on the reviewers suggestion, we attempted to add that to the summary figure or as a supplemental. Given that the available density data varies significantly in terms of surveillance sampling effort, frequency and geographic region, the information did not map well in a way that informed on the three different species. We have instead added an additional bar chart as part (b) of Figure 5 that bins the available data according to region of the Atlantic and helps to provide a visual representation of species density in the broad regions of interest. This is intentionally placed under the map to encourage easier interpretation of the east to west alignment.

Referee: 2

Comments to the Author(s)

This paper examines the phenomenon of phocine distemper virus outbreaks in Atlantic pinniped populations, which have caused significant mortality in some cases and received a lot of scientific and public interest over the past decades. One of the main conundrums about these outbreaks is why they appear to be limited to the eastern Atlantic, whereas outbreaks of comparable severity have not been seen in populations of the same species on the western side of the Atlantic. The study seeks to provide new answers to this question through a combination of viral genetic and seal host serology data. Based on these data, the author argue that continuous circulation of a less pathogenic form of the virus in the northwest Atlantic might maintain high enough levels of immunity with these populations to prevent the kind of outbreaks seen in eastern populations.

I found the study interesting and well presented but I am not convinced by the authors' data and conclusions. Specifically, the argument about endemic circulation creating partial herd immunity rests on such endemic viruses being limited to western Atlantic populations but being absent in the East. Published data to document this might exist but if so they should be presented for comparison. The part about genetic differentiation of the viruses on either side of the Atlantic is interesting but doesn't appear to be novel. The last part of the study, focused on variation in the virus' hemagglutinin gene which the authors argue is responsible for different virus phenotypes, seems very speculative and likely involves incorrect analyses (see below). I appreciate that the data for this type of work are hard to come by and that the current study is exceptional in terms of the amount of data that has been assembled. Still, the study would benefit from strengthening the arguments where possible but to otherwise refrain from speculative claims and present the findings as hypothesis-generating rather than confirmatory. There are also a several methodological issues that need to be addressed.

Given the speculative nature of the possible cross-protection from a putative less virulent strain, we have reworked the full manuscript to decrease that discussion point, and refocused primarily on the ongoing presence of PDV in North America. The interpretations on the regional variant have been reworded to more clearly convey that they are meant to be hypothesis generating

and intriguing considerations for future work. We have also addressed the methodological concerns that were raised, with each further described in the specific points below.

Specific points:

1. Based on serological and qPCR results, the authors argue that some form of PDV is circulating endemically in North American seals. For comparison, do we know that such evidence of consistent exposure is definitely not detectable in European populations? The authors cite data from the eastern Atlantic comparable to their own in the discussion but for the readers to evaluate this comparison it would be helpful to include these published data in their figures or tables. It is also not clear to me whether the available data are limited to serology or whether the same patterns hold when screening eastern populations by qPCR for circulating PDV.

There have been a handful of published studies looking for PDV in European populations. Cumulatively they have included over one thousand animals, primarily harbor seals, though some grey and ringed seals have also been reported. Published surveillance includes samples from 1988 through 2014, spanning the first European UME and extending through 12 years beyond the second UME. Studies have focused primarily on serology, though a smaller dataset of 117 animals from the North Sea and Greenland did include RT-PCR screening and failed to detect any RT-PCR positive animals. From the serology data in Europe, antibodies are detectable for the first few years post UME, but rapidly decline to a point of being undetectable except in the case of older animals who had lived through one of the UME timeframes. Beyond 2 years post UME, antibodies are no longer detected in young of the year or pups. This point has been elaborated on with citations in the introduction text of the manuscript (lines 66-70).

2. Please indicate which sequences in Fig 1 were new and which had been previously published and analysis. This would make it easier to appreciate what the current study is able to add to previous work and what had been known before in terms of geographical clades and their divergence times. It would also be helpful to include highest posterior density intervals on the key internal nodes (i.e. split between NE and NW Atlantic).

The manuscript has undergone significant restructuring in order to make the data presentation flow more clearly. As such the previous Figure 1 is now Figure 2 in the current version. In all text, figures, and tables throughout the manuscript all newly reported sequences are now marked with an asterisk. Each data presentation also includes more detailed description of which sequences were included. The split between NE and NW Atlantic is estimated at 2001 (HPD 95% 1998-2002) and has been added in the results (line 295-297).

3. Before attempting to reconstruct time-scaled phylogenies it would be important to confirm that there is actually enough temporary signal (increase in divergence over time) to estimate a molecular clock. This should be done based on the ML phylogeny using TempEst and results could go into the supplement rather than the main text.

&

4. Results of the model selection for molecular clock and demographic priors should be documented - the authors state that they decided on an exponential growth model but we don't know what other models were considered. Exponential growth might not be an obvious choice

for a virus that appears to be circulating endemically and the choice of demographic prior can have a significant effect on estimated divergence times.

These analyses are now described in the supplemental materials methods with supporting figures in supplemental Figure S1, supplemental table S2 and noted in the main text in lines 200-202.

The H gene tree reconstructed using maximum likelihood methods with dated tips was input into TempEst v1.5.3 to evaluate clock-like evolution. The results indicate that divergence increased as a function of time with an R-squared value of 0.859, and a correlation co-efficient of 0.9273. To perform this analysis the tree was evaluated using the 'best-fitting root'. We interpreted this as a good temporal signature, indicating that the H gene of PDV is fit for analysis with Bayesian phylogenetics (Figure S1).

A table describing the model testing output has been added to the supplemental material as Table S2 and is now described in the supplemental methods. All raw data and output from model testing in BEAST have been added to the github repository (https://github.com/ksawatzki/Supp_data/). Six parameterizations were tested using path and stepping stone sampling in BEAST, and the log Bayes factor compared. Constant size models using strict and relaxed clocks were used as the null, and compared against exponential growth and GMRF Bayesian Skyride models with strict and relaxed clocks.

5. The selection analysis in Table S4 looks wrong: the simpler models (M1a, M7) have a much higher likelihood than their complex counterparts with two extra parameters, which shouldn't be the case (additional parameters might fail to improve the likelihood but they can't reduce it). This suggests that some of the PAML analyses didn't converge and that the results are unreliable. Given that there appears to be a lack of sites experiencing more than one non-synonymous substitution, I would be surprised if the data contained any evidence of positive selection. I don't think it is appropriate to use the BEAST tree as the topology for the PAML analysis, this should be a tree estimated without a molecular clock.

We appreciate this useful interpretation of the PAML output and we agree with the reviewer. We have decided to remove this from the manuscript.

6. L378-382. I don't understand the point made in this paragraph - genetic variation was found in this position, but how is this evidence of intra-host selection?

This section has been removed from the text.

7. L391-393. The effect of these mutations on the virus ability for fusion is complete speculation - it is not appropriate to refer to this as a 'low-fusion lineage' in the Discussion

This has been either completely removed, or significantly softened and reworded as speculative in the few instances where the reference was preserved.

minor comments:

- description of virus genomic sequencing is not consistent (e.g. read length, paired/unpaired reads). Please add missing information

We agree that the genomic sequencing methodology was confusing as presented. The 6 genomes were generated at 3 different institutions using different platforms and processing approaches, even within an institution. We have reorganized and reworded the methods so that details are provided for each genome, rather than by institution and the information is presented in a more consistent manner. More specific sequencing methods were added where missing.

- line 430 - I believe this should say "Northeast Atlantic"?

The manuscript has been significantly reworked and this specific line no longer exists – we have carefully reviewed the current version for similar mistakes.

- the scale bars on the phylogenetic trees represent the 'substitution rate' not 'mutation rate' - please change throughout

This has been corrected throughout the manuscript.

- I like Fig. 5 but I find it very difficult to read, especially the part under the time line in the right panel

We agree that the annotations under the figure were difficult to read. We have removed them from the figure as we have decided that they were not necessary to the purpose of that figure and created unnecessary clutter. Figure 5 has been further modified to include a 3rd panel (now Figure 5b) to show species density.

- Highlight species of interest in the trees shown in supplemental figures S2 and S3?

This has been added. Grey and harbor seals are marked in blue, other marine mammals are marked in yellow.

- how was the tree in Fig 3 rooted?

The previous unrooted RAxML Figure 3 has been removed and is now a time scaled phylogenetic analysis of the H gene.

- the trees would benefit from better formatting: if the viewer is supposed to evaluate the position of specific taxa (e.g. 'orphan sequences', sequences from UME or non-UME events, different host species), these should be visually highlighted in the tree

This has been fixed throughout. Species are now defined by color and newly reported sequences are all marked with asterisk. Sample names have been standardized and simplified where possible.

Appendix B

In regard to the two specific suggestions highlighted by the editor:

- (1) we have uploaded the serological titer data to our electronic supplementary material which is now publicly available (https://github.com/ksawatzki/Supp_data/) and
- (2) we have moved the bulk of sequence generation methods to the Supplemental material document.

Reviewer comments are in black

Author responses are in blue

Changes to the text are in red

Referee: 2

The authors did a commendable job incorporating the reviewers' comments and have made it much more clear what the contribution of their study is and what their key findings are. I only have a few specific points, listed below, which I suggest the authors should look.

One general point is that the authors could do more to convey the broader importance of their work. The abstract for example is quite focussed on the virus studied but provides little indication as to why we need to know these things and what can be taken away from the study. Instead of reporting in the final sentence what sequence data have been generated, I would expect to see some general conclusions, possibly beyond PDV. Similar opportunities exist in the Introduction and the Discussion.

We have added language to both the abstract and the discussion for how this work helps to provide a framework for understanding disease transmission in shifting ecologies and populations. Specifically:

Lines 46-48: Work presented here provides a framework toward greater understanding of how recovering populations and shifting species may impact disease transmission.

Lines 400-406: Lastly, we identified a regional variant that we hypothesize may provide some population level protection that limits the severity of outbreaks and could be useful in future modeling efforts evaluating the impact of PDV. High-latitude ecosystems are expected to be increasingly disrupted by climate change, altering the conditions for disease ecology with multiple stressors. The observations reported here help to provide a framework for emerging disease as ecological conditions and species distribution and overlap continue to shift.

specific points:

I appreciate the additional information provided about the marginal likelihood estimation and model comparison in BEAST in the supplement. Were those done using default settings? If so, please state that or otherwise provide further detail. Some of the results look questionable - the two methods (stepping stone and path sampling) should produce very similar estimates but for some of the models, including for the model selected as the top one, there are major differences, in the order of several hundred log units. This suggests that these analyses were probably not run for a sufficient length of time (steps). I don't expect this change the main conclusions of the paper but it does not look like good practice.

All models were run with specified tree priors and clocks using default parameters for the HKY substitution model. This has been clarified in the methods. We further thank the reviewer for noticing the PS/SS difference in two tested models. In light of this, we re-ran these model tests using longer chain lengths, which resulted in highly comparable PS/SS log marginal likelihood values which have been amended to Supplemental table 2. This resulted in the constant model having slightly better support than exponential growth, and therefore we re-ran the BEAST analysis using the constant size for tree prior. As anticipated, the output are qualitatively indistinguishable, with only within-clade movement in homogenous clusters. We have replaced figures 2 and 3 with the result of the updated model. There is no change in quantitative results (reported tMRCAs and 95% HPD), our interpretation or text. We sincerely appreciate the expert guidance in strengthening the support for this paper.

It is not clear how the authors arrived at the "estimated minimum level of 250,000 needed to support endemic infection of morbillivirus" (line 1108-1110). This is a really interesting point (mirroring what is seen in other morbilliviruses like measles, maybe cite some of this work?). The cited reports look like they would contain estimated seal numbers but did they also report this persistence threshold for PDV? If so, how was it inferred?

As the reviewer has noted, this is the estimate for morbillivirus that has been derived from measles and the references were inadvertently omitted. We thank the reviewer for catching this omission and have corrected it. These references are citations 39-41 (Black, Bartlett, and Keeling & Grenfell).

line 545 - poorly constructed sentence: "Individual PDV genes were aligned for... genomes..." - why not simply "Sequences were aligned using Clustal"?

This change has been made as suggested:

Line 173: Sequences were aligned using ClustalΩ (v1.2.4).

line 550 - how were dates formatted when only year was available? Or where all dates simplified to 'year only'?

For BEAST tip dating, in the few cases when only a year was available, the tip date was set to the midpoint of the known time period (year or month) with uncertainty set to cover the full period. For instance, if only a year was known, tip date was set to 'YYYY.5' with 0.5 uncertainty. We have included a sentence describing this in the methods. For figures, dates were simplified to year only in response to a prior reviewer suggestion to simplify the sequence names throughout the manuscript.

Line 178-180: When exact date was not known, the date was set at the mid-point of the known month or year with uncertainty spanning the time period.

line 554 - should say "RELAXED CLOCK with an uncorrelated lognormal distribution"

This change has been made as suggested.

Line 181-184: Final analyses were independently run 5 times on BEAST (v1.10.4) with a chain length of 100 million generations using a constant growth coalescent tree prior, relaxed clocked with an uncorrelated lognormal distribution, an HKY γ substitution model, and default parameters

line 798 - I wonder about the heading: to me the key finding here is not the single introduction (there might have been others that simply weren't detected) but the continuous circulation of a single PDV lineage in the Northeastern Atlantic since the early 2000's

This heading has been changed.

Line 262: Continuous circulation of a single PDV lineage in the Northwest Atlantic

line 816, 997, 1128 - 'predicted' should be replaced with 'hypothesised'. The word prediction has a specific meaning in science and generally involves some quantitative (i.e. statistical) basis so it doesn't seem appropriate here.

The changes have been made as suggested.

Line 279-280: A clade seen in regional pinnipeds in eastern North America exhibits a Hemagglutinin substitution hotspot hypothesized to decrease fusogenic activity

Lines 306-308: Given the location of these substitutions in a region critical for viral fusion and the persistence of this substitution over multiple years, we hypothesize viruses with this hotspot may have impaired fusogenicity.

Lines 381-382: This regional variant has a persistent substitution hotspot that we hypothesize will decrease viral fusion and may result in a naturally attenuated virus.

line 825 - "...and is referred to ..." should be "... with the latter being referred to..."

This change has been made as suggested.

Lines 88-90: All sequences from the Northwest Atlantic spanning 2011-2015 fell into one conserved, distinct lineage that was not shared by any of the Northeast Atlantic sequences with the latter being referred to as the North American lineage from here on.

Fig 3 would benefit from some additional formatting to help the reader. For example, it would be helpful to distinguish the Northwestern and Northeastern sequences as in Fig. 2. The colour used on indicate harp seals looks different from the one in Fig 2.

The color used in Figure 3 has been adjusted to match the color used in Figure 2, and the sequences have been labelled to distinguish the Northwestern and Northeastern sequences.

Appendix C

Attn: Editorial Board, *Proceedings of the Royal Society B*, October 18, 2021

Re: Cover letter for revision of accepted research article

Dear Dr. Heesterbeek,

We have revised the data repository readme file as requested:

https://github.com/ksawatzki/Supp_data/blob/main/PDV_ProcRoyalSocB/Readme.txt

I have included the text below for your convenience. No other changes have been made.

Thank you, we are excited the paper has been accepted and look forward to its publication.

Sincerely,

Kaitlin M. Sawatzki and Wendy Puryear, on behalf of all co-authors

Readme.txt

Data and analysis for

"Longitudinal analysis of pinnipeds in the Northwest Atlantic provides insights on endemic circulation of Phocine distemper virus"

Wendy Puryear, Kaitlin Sawatzki, Andrea Bogomolni, Nichola Hill, Alexa Foss, Iben Stokholm, Morten Tange Olsen, Ole Nielsen, Thomas Waltzek, Tracey Goldstein, Kuttichantran Subramaniam, Thais Carneiro Santos Rodrigues, Manjunatha Belaganahalli, Lynda Doughty, Lisa Becker, Ashley Stokes, Misty Niemeyer, Allison Tuttle, Tracy Romano, Mainity Batista Linhares, Deborah Fauquier, Jonathan Runstadler

Proceedings of the Royal Society B: Biological Sciences

DOI: 10.1098/rspb

If you have questions, please contact:

Kaitlin Sawatzki, kaitlin.sawatzki@tufts.edu - technical questions, data access

Wendy Puryear, wendy.puryear@tufts.edu - data collection, permitting

Description of data

Alignments directory includes all sequences used in the manuscript and supplement, aligned and in FASTA format that were read into BEAST and RAxML for phylogenetic analysis.

PDV sequences are in nucleotide format. Nectin-4 and SLAM are amino acid. GenBank accession numbers are as follows: Full genome - MW642077 (US/Hg/IFAW18-001/2018), MW642078 (US/Pv/NEAQ18-044/2018), MW504062 (CA/Pg/WVL181140/2017), MW581015 (US/Pv/MME-287/2018), MW581016 (US/Pv/MME-343/2018), MW581017 (US/Pv/MME-430/2018); Hemagglutinin - MW581018- MW581026
Nectin-4 and SLAM Refseq accessions are included in a tab-delimited text file in their directory

BEAST directory includes the following input and output files from BEAST runs for PDV full length, partial genome and HA gene analyses:

BEAUTi-generated input file with all run parameters and metadata
BEAST-generated log file output
BEAST-generated tree file output
Combined tree runs and maximum clade credibility tree files
BEAST directory includes the following input and output files from BEAST model selection:

BEAUTi-generated input file with all run parameters and metadata
BEAST-generated log file output
MLE output files

RAxML

RAxML log output file with all run parameters and metadata
RAxML best scoring ML tree files

Seal_data

Data used to make figures 1 and 5
Serology titers and metadata